Methods

# A scalable approach to topic modelling in single-cell data by approximate pseudobulk projection

Sishir Subedi[1,2], Tomokazu S Sumida[3], Yongjin P Park[2,4,5]

Probabilistic topic modelling has become essential in many types of single-cell data analysis. Based on probabilistic topic assignments in each cell, we identify the latent representation of cellular states. A dictionary matrix, consisting of topic-specific gene frequency vectors, provides interpretable bases to be compared with known cell type–specific marker genes and other pathway annotations. However, fitting a topic model on a large number of cells would require heavy computational resources–specialized computing units, computing time and memory. Here, we present a scalable approximation method customized for single-cell RNA-seq data analysis, termed ASAP, short for Annotating a Single-cell data matrix by Approximate Pseudobulk estimation. Our approach is more accurate than existing methods but requires orders of magnitude less computing time, leaving much lower memory consumption. We also show that our approach is widely applicable for atlas-scale data analysis; our method seamlessly integrates single-cell and bulk data in joint analysis, not requiring additional preprocessing or feature selection steps.

## Introduction

### Background

High-throughput single-cell sequencing has gained popularity and has been successfully applied to many recent studies to understand cellular contexts. Not only the number of studies that involve single-cell sequencing but also the number of cells that a single study can profile has dramatically increased from 100s to millions. Advancements in computational methods have necessarily accompanied such technological advances in order to provide necessary operations in almost every step of data analysis—from data procurement to final statistical inference (Kharchenko, 2021). However, typical computing infrastructures are often not well equipped with powerful computing units and high-capacity memory despite the fact that many advancements in computational methods assume such large resources are readily accessible (Heumos et al, 2023; Zhang et al, 2024). Training a sophisticated model on millions of single-cell data vectors, using a stochastic gradient descent algorithm, not surprisingly demands much more resources than fitting a model on data with 1,000s of cells and a 1,000 features.

After the quantification of molecular features within each cell, a conventional single-cell analysis (Stoeckius et al, 2017; Stuart & Satija, 2019; Hao et al, 2021) generally turns to dimensionality reduction, such as principal component analysis (PCA) (Hotelling, 1933; Jolliffe, 1986), nearest neighbour graphs were constructed by matching cells based on the reduced data, and cell clusters are often resolved by a graph-based clustering method, such as the Louvain (Blondel et al, 2008) or the Leiden algorithm (Waltman and van Eck, 2013; Traag et al, 2018 Preprint). The complexity of an exact PCA algorithm (singular value decomposition) scales linearly with the number of genes/features and quadratically with the number of cells (Tsuyuzaki et al, 2020), designing a scalable latent data modelling is crucially important (Sun et al, 2019). Should a large number of cells be analysed, finding hidden gene regulatory programs/factors/topics is a computationally intensive task. It is customary to conduct an unvetted feature selection for highly variable genes to reduce memory footprint. By doing so, we critically assume that biologically relevant signals are most pronounced in data and further take that the rank of gene expression variance of the selected genes is somewhat stable across different data sets.

### Related work

How can we uncover latent gene programs without compromising the intrinsic dimensionality? Several approaches have been suggested to overcome the scalability issues of the embedding or dimensionality reduction step before or after building neighbour graphs, and the ultimate outcome is to resolve clusters of cells using a graph-based clustering method. Most existing methods address the question of how we select fewer representative cells to ease the computational burden of downstream analysis, not sacrificing the accuracy of the final clustering results (Hie et al, 2019b;

[1]Bioinformatics Graduate Program, University of British Columbia, Vancouver, Canada   [2]BC Cancer Research, Vancouver, Canada   [3]Neurology, Program for Neuroinflammation, Yale School of Medicine, New Haven, CT, USA   [4]Department of Pathology and Laboratory Medicine, University of British Columbia, Vancouver, Canada   [5]Department of Statistics, University of British Columbia, Vancouver, Canada

Correspondence: yongjin.park@ubc.ca

Abdelaal et al, 2020; Dhapola et al, 2022; Wei et al, 2022). A geometric sketching method was first coined to randomly sample cells while uniformly covering an underlying and unknown metric space (Hie et al, 2019b). A similar idea was explored in natural language processing but more focused on defining the convex hull (boundary) of word occurrence topics over document corpus (Mimno & Lee, 2014). More recent scalable approaches developed by the genomics community also build on the same premise that we can approximately capture the overall clustering patterns across millions of cells by choosing representative (anchor) cells wisely (Abdelaal et al, 2020; Dhapola et al, 2022; Wei et al, 2022). After building nearest neighbour graphs of the anchor cells, the rest of the algorithmic steps in all these approaches largely focus on adding new cells (vertices) to the original nearest neighbour graphs of the initial anchor cells and performing label propagation of cell-type annotations.

### Our approach

Instead, here, we focus on developing a scalable approach for probabilistic topic modelling. We deal with high dimensionality and large sample size by mapping high-dimensional expression/activity vectors onto the lower dimensional topic space via random projection. Probabilistic topic modelling decomposes a large count data matrix into an interpretable dictionary matrix (topic-specific activities across 10s of 1,000s of genes/features) and a topic proportion matrix (each cell's attribution to the topics) (Blei et al, 2003; Dieng et al, 2020). Large-scale single-cell analysis can be made quite rapidly by considering the data generation process from a rather different perspective. Instead of designing a new type of complex model while leaving its statistical inference up to machine-learning libraries, we propose a new framework in which a fundamental topic model can be estimated in an accurate, robust, and scalable way.

### Why topic modelling

Fitting a probabilistic topic model has long been considered a principled and intuitive approach to uncovering patterns hidden underneath count data derived from high-throughput sequencing. A grade of membership model (or admixture) was first coined in genetics while trying to identify population structures manifested in genetics variant counts across the genome (Pritchard et al, 2000; Novembre, 2016). In the same way, multi-tissue, multi-individual gene expression patterns were analysed by the same type of model (Dey et al, 2017). More recently, the Poisson matrix factorization (PMF) was shown to be an equivalent problem, and a more efficient method based on alternating regression estimation was suggested (Carbonetto et al, 2021 Preprint, 2023 Preprint). In single-cell genomics, especially for sparse DNA accessibility data, embedding methods based on topic modelling (or latent semantic indexing), such as ArchR (Granja et al, 2021) and cisTopic (Bravo González-Blas et al, 2019), were ranked in the top lists of a recent benchmark study (Chen et al, 2019). A topic model approach based on a deep variational autoencoder model, called an embedded topic model, or ETM (Dieng et al, 2020), has been successfully used in single-cell RNA-seq modelling (Zhao et al, 2021b; Subedi & Park, 2023; Zhang et al, 2023).

### Our work

We present a new scalable and versatile method that can quickly recover cellular topics with modest computing resources. We term our method ASAP, short for Annotating a Single-cell data matrix by Approximate Pseudobulk estimation. Several existing works and observations inspired us. First, repeatedly applying random projection operation onto high-dimensional feature vectors may result in quick and moderately accurate cell clustering patterns (Wan et al, 2020). Second, pseudobulk data derived from aggregating within cell types and states often behave similar to typical RNA-seq bulk data. Third, a grade of membership, or an equivalent non-negative matrix factorization (NMF), method is powerful enough to dissect cellular topics from bulk sequencing profiles (Dey et al, 2017) and single-cell sequencing data (Carbonetto et al, 2021 Preprint).

# Results

### Overview of ASAP

Briefly, ASAP will handle a matrix factorization problem of massive single-cell data in the following three steps: (1) we randomly project cells onto the low-dimensional space (Fig 1A) to sort them through a binary classification tree (Fig 1B), depending on the sign of the random projection (RP) values, and construct a pseudobulk (PB) data matrix by aggregating cells landed in the same termini of the binary sorting tree. As suggested by the previous work (Wan et al, 2020) and demonstrated in the UMAP (Becht et al, 2018) (Uniform Manifold Approximation and Projection), simple RP operations can help cells with a similar expression variation group together (Fig 1B). It is also encouraging to find that the resulting PB data already show that cells of similar cell types are naturally enriched within the same PB sample (the structure plot of Fig 1B, cell-type proportion by PB samples). (2) Followed by the NMF, or, more precisely, the PMF, we can decompose the PB expression data ($Y_{pb}$) into topic-specific gene expression (dictionary) matrix ($\beta$) and topic loadings/proportions for each PB sample ($\theta_{pb}$). The non-negativity constraint is natural to the gene expression count data, and the additivity of factors generates biologically interpretable solutions. (3) Treating the dictionary matrix as a design matrix in a non-linear, non-negative regression problem, we can quickly deconvolve cell-level topic proportions (Fig 1D). The resulting topic proportion matrix often results in markedly improved UMAP results (Fig 1B versus Fig 1D). Moreover, the dictionary matrix $\beta$ and two topic proportion matrices $\theta$, estimated by the PMF algorithm (Gopalan et al, 2014, 2016; Levitin et al, 2019), clearly exhibit modular structures in both sides (see the Materials and Methods section for technical details).

### ASAP accurately estimates cell clusters

In single-cell data analysis, it is implicitly assumed that we collect a sufficient number of cells within each cell type, and these cells show more similar expression patterns within the same cell type than between different cell types. Some intrinsic, biological

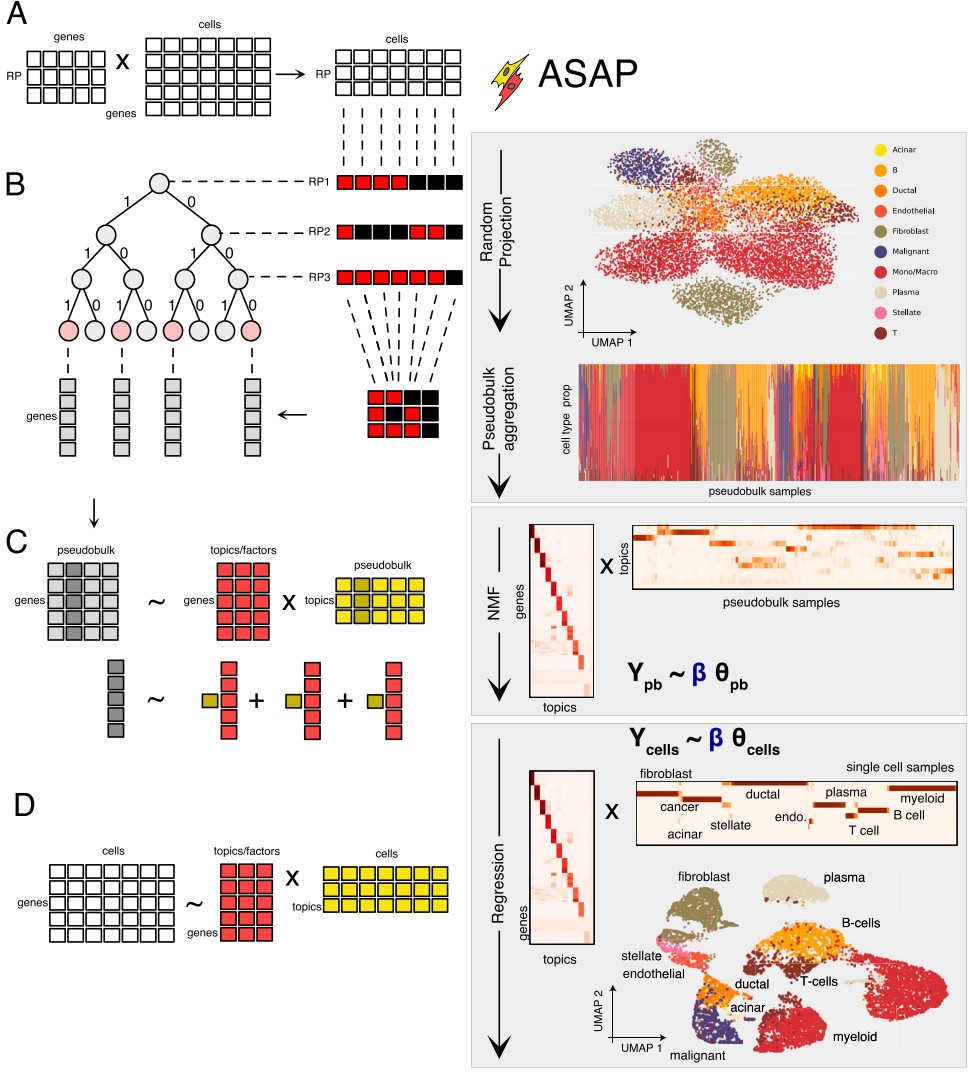

**Figure 1. Schematic overview of the ASAP framework.**
**(A, C, D)** ASAP proceeds in three steps: collapsing cell count data into a manageable pseudobulk matrix (A), factorizing the pseudobulk matrix by the Poisson matrix factorization (C), and regressing cell-level data onto the topic space (D). For illustration purpose, we show intermediate results in processing pancreatic ductal adenocarcinoma data, consisting of 9,887 cells (Zhao et al, 2021a). **(A)** Raw gene expression matrix can be projected on the *r*-dimensional space of RP variables (right) by multiplying the observed cell count matrix (middle) with some univariate Gaussian matrix (left). **(B)** According to the sign of each random projection (RP) direction, switching to the left and right at each level, cells are sorted via a perfect binary tree with the depth =r (RP variables) and aggregated into pseudobulk samples. **(C)** Non-negative matrix factorization method decomposes the pseudobulk data matrix into a dictionary (gene by topic/factor) and factor loading matrix (sample by topic/factor). We represent each pseudobulk sample as a non-negative linear combination of non-negative topic-specific gene expression topics. **(D)** Treating the dictionary matrix as a design matrix, ASAP revisits the original gene expression vectors of many cells and regresses them on the topic space with the same non-negative constraints on the regression coefficients, which can be used to identify cell-level clustering patterns (Uniform Manifold Approximation and Projection on the right).

similarity measures can group cells of identical cell types/states. Had dimensionality reduction or latent embedding methods well preserved cells' biological similarity, clustering results based on the latent factors/topics should have been closely matched with the groups based on ground truth cell types. Hence, we evaluate the performance of different embedding methods in terms of the correspondence between cell groups found by clustering each method's latent factors/topics and ground truth cell types.

We generated realistic simulation data treating cell type–specific gene expression profiles in cell sorting experiments (Schmiedel et al, 2018). We used the sorted bulk sequencing data instead of other single-cell data sets that are computationally annotated by some clustering methods. To gain insights into variability induced by sequencing depths and sample size, we varied the number of cells from 1,000 to 10,000, while also varying the proportion of cell type–specific gene expression and background variation (see the Materials and Methods section). Because we are interested in comparing the estimated set membership (based on clustering) with actual cell-type membership, we used three performance metrics widely used to evaluate clustering tasks: adjusted Rand index (Rand, 1971), normalized mutual information (Vinh et al, 2010), and purity score (Liu et al, 2020; Kriebel & Welch, 2022).

## Random projection well preserves underlying cell-type identities

We first wanted to understand the general behaviour of random projection operation. A similar analysis was conducted in previous studies, such as Wan et al (2020). As we increase the purity (ρ) level (Fig 2A), patterns emerge, and cells form groups (Fig 2B). Because visual inspection via UMAP is not rigorous enough, we evaluated the accuracy of the Leiden clustering results (Traag et al, 2018 *Preprint*) based on different configurations of latent factors derived from PCA and RP with a different number of components/factors (Fig 2E). Although the performance of RP data generally lags behind that of PCA by significant statistical margins, noting that PCA is more computationally intensive, we found that RP operation can be used as a good-enough-approximate method for the subsequent steps. Conversely, it also shows that RP alone is insufficient to capture a

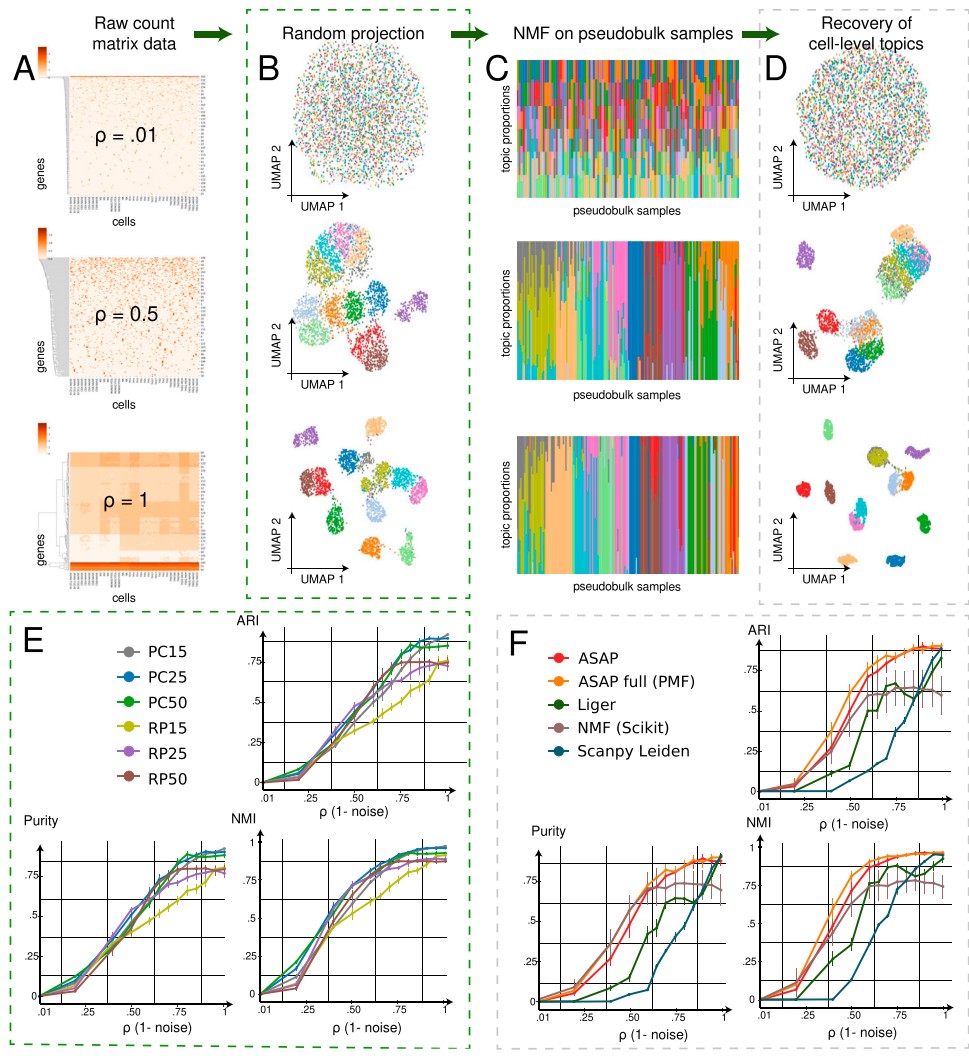

**Figure 2. ASAP accurately estimates cell clusters.**

Extensive benchmark experiments confirm that our approximating method performs better or at least as well as existing methods. **(A)** We generated realistic simulation data treating cell type–specific gene expression profiles in cell sorting experiments (Schmiedel et al, 2018) as a gold standard while applying a different noise level, or conversely increasing the proportion of cell type–specific variance from $\rho$ = .01 to $\rho$ = 1 (the top: $\rho$ = 0.01; the middle: $\rho$ = 0.5; the bottom: $\rho$ = 1). **(B)** We can reduce dimension by multiplying the simulation count matrix with a random Gaussian matrix (see Fig 1A and Materials and Methods section). Uniform Manifold Approximation and Projection visualization shows how cell type–specific clusters emerge as we increase the proportion of cell-type variance from the top ($\rho$ = 0.01) to bottom ($\rho$ = 1). Each dot represents a cell with its originating cell type (colour). **(C)** Poisson matrix factorization decomposes a pseudobulk data matrix derived from the previous random projection step, yielding a topic-specific gene/feature frequency matrix ($\beta$) and topic proportions of each sample ($\theta$). Here, we show the topic proportion matrices ($\theta$), where each colour bar (x-axis) represents each sample's probabilities of topics (y-axis; colours), and they sum to one. **(D)** Cell-level topic regression refines cell type–specific clusters. We show the Uniform Manifold Approximation and Projection based on the final topic regression results. **(E)** We tested the impact of different choices of RP dimensions on cell-type clustering performance. We compared clustering results derived from different embedding methods with the gold standard cell-type annotations (cell-sorted data). We report three different metrics on the y-axes—ARI (adjusted Rand index, the left), NMI (normalized mutual information, the middle), and purity (the right) while varying the noise levels. For all the metrics, the higher, the better. RP-$k$: the Leiden clustering of cell–cell 15-nearest neighbour graphs built on $k$-dimensional random projection results. PC-$k$: the Leiden clustering of cell–cell 15-nearest neighbour graphs built on top $k$ principal components. **(F)** We compared the accuracy of ASAP's final topic embedding results against other non-negative matrix factorization (NMF)–based methods, such as ASAP-full (the Poisson factorization directly applied to full data), Liger (Welch et al, 2019; Kriebel & Welch, 2022), NMF function, sklearn.decomposition.NMF, implemented in the scikit-learn library (Pedregosa et al, 2011), and a standard clustering pipeline implemented in the Scanpy library (Wolf et al, 2018). X-axis: the proportion of cell-type variance; y-axes: ARI (adjusted Rand index, the left), NMI (normalized mutual information, the middle), and purity (the right); for all the metrics, the higher, the better.

detailed view of cellular diversity, but the results of RP need to be refined.

## PMF methods achieve a higher level of robustness

### Pseudobulk data

As a refinement step, we used the PMF (Gopalan et al, 2014, 2016; Levitin et al, 2019) applied to the PB data matrix (ASAP) or the full data matrix (ASAP-full/PMF). The PMF will decompose the resulting non-negative PB matrix into topic signatures ($\beta$) and topic proportion profiles ($\theta$, Fig 2C). So, it was natural to compare with other NMF-based methods (Fig 2F), which include Liger (Welch et al, 2019; Kriebel & Welch, 2022), a standard NMF method (sklearn.decomposition.NMF) in the scikit-learn library (Pedregosa et al, 2011), along with a conventional Leiden clustering pipeline implemented

in the Scanpy library (Wolf et al, 2018). All the methods were summarized into the clustering results, and cell memberships were tested against the gold standard (see the Materials and Methods section for details). Conceptually, we expect good clustering results distinctive clouds of points/cells in the topic space as visualized in UMAP (Fig 2D). The ASAP, Liger, and Scanpy's clustering method naturally incorporates the Leiden clustering as a final step. For the Leiden-based methods, we varied the resolution parameters (0.1, 0.25, 0.5, 0.75, and 1.0); our results were, albeit, invariant to the fine-tuning of the Leiden algorithm. For the scikit-learn NMF, we simply used the standard k-means algorithm implemented in the same library.

We found that the PB-based ASAP and ASAP-full clearly outperformed other NMF and clustering methods with statistically significant margins, especially in settings with moderate and high

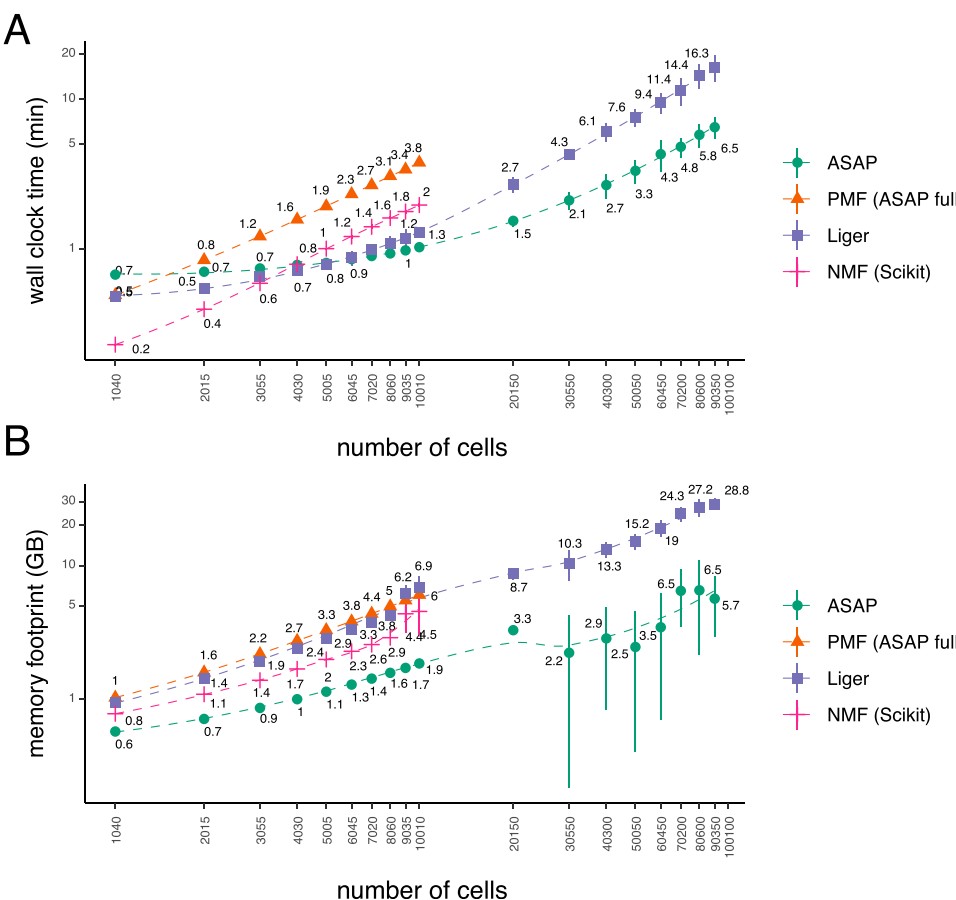

**Figure 3. ASAP achieves scalability in terms of computation time and memory footprint.**
**(A, B)** Plot shows (A) the elapsed time (in minutes) and (B) peak memory usage (in GB) by non-negative matrix factorization–based methods with different numbers of cells ranging from 1,000 to 100,000 as a function of the number of cells. For the cases with the number of cells exceeding 10,000, we focused on comparisons between ASAP and Liger because our computing resource was insufficient to conduct full matrix factorization with a full set of features using non-negative matrix factorization implemented in the scikit-learn (Pedregosa et al, 2011) and the Poisson matrix factorization (Gopalan et al, 2014, 2016; Levitin et al, 2019) (ASAP-full).

noise levels (Fig 2F), suggesting that the PMF algorithm, which both ASAP and ASAP-full were built on, is more robust than other NMF algorithms. The PMF is a Bayesian hierarchical model, resolves its parameters by posterior inference, and often yields more robust results than optimization-based algorithms. Perhaps a more important aspect of this benchmark result is that our ASAP method based on PB data is as competent as the PMF fitted on full data (the red and orange curves in Fig 2F).

## ASAP is a scalable method with little memory footprint

Along with the performance metrics (Fig 2), we also measured the runtime (in minutes) and memory footprint (in gigabytes/GB) of each method in the same workstation. We allocated the same resources for all the experiments, fixing the available memory to 32 GB and eight CPU cores. As long as the allocated memory permits, we varied the number of cells from 1,000 to 100,000. In the first batch of experiments, varying the number of cells from 1,000 to 10,000, ASAP-full (vanilla PMF) often took the most time (Fig 3A) and emended high memory to keep track of multiple copies of variational parameters (Fig 3B), followed by Liger and scikit-learn NMF methods. However, ASAP was consistently fast enough to get the results within a minute and kept the peak memory usage under 2 GB until the 10k cell experiments.

In the second batch, we focused on Liger and ASAP models, varying the number of cells from 20,000 to 100,000. Here, ASAP showed significantly faster runtime and lower memory usage. Liger already peaked its memory usage near 32 GB with 100,000 cells. Therefore, it was not included in further experiments. We also conducted further experiments with ASAP alone to measure runtime and memory usage and found them to scale linearly with the number of cells (Supplementary Figure). Regretfully, our computing resources were limited; we had to focus on comparisons between the ASAP and Liger for experiments with the number of cells exceeding 10,000. A full matrix factorization with a full set of features using NMF implemented in the scikit-learn (Pedregosa et al, 2011) and PMF (ASAP-full) required substantially large memory (>128 GB) while taking days of computing time.

## ASAP recapitulates known cell type–specific gene programs in real-world data sets

We applied ASAP to three different large-scale scRNA-seq data sets to demonstrate that our method is scalable and accurate enough to recapitulate well-established cell type–specific gene expression patterns. We considered the following data sets as representative examples: (1) PBMCs (Hao et al, 2021) profiled on 161,764 cells and 20,729 genes (Fig 4A–C); (2) breast cancer mammary tissues (Wu et al, 2021) of 20,265 genes expressed on 100,064 cells (Fig 4D and E);

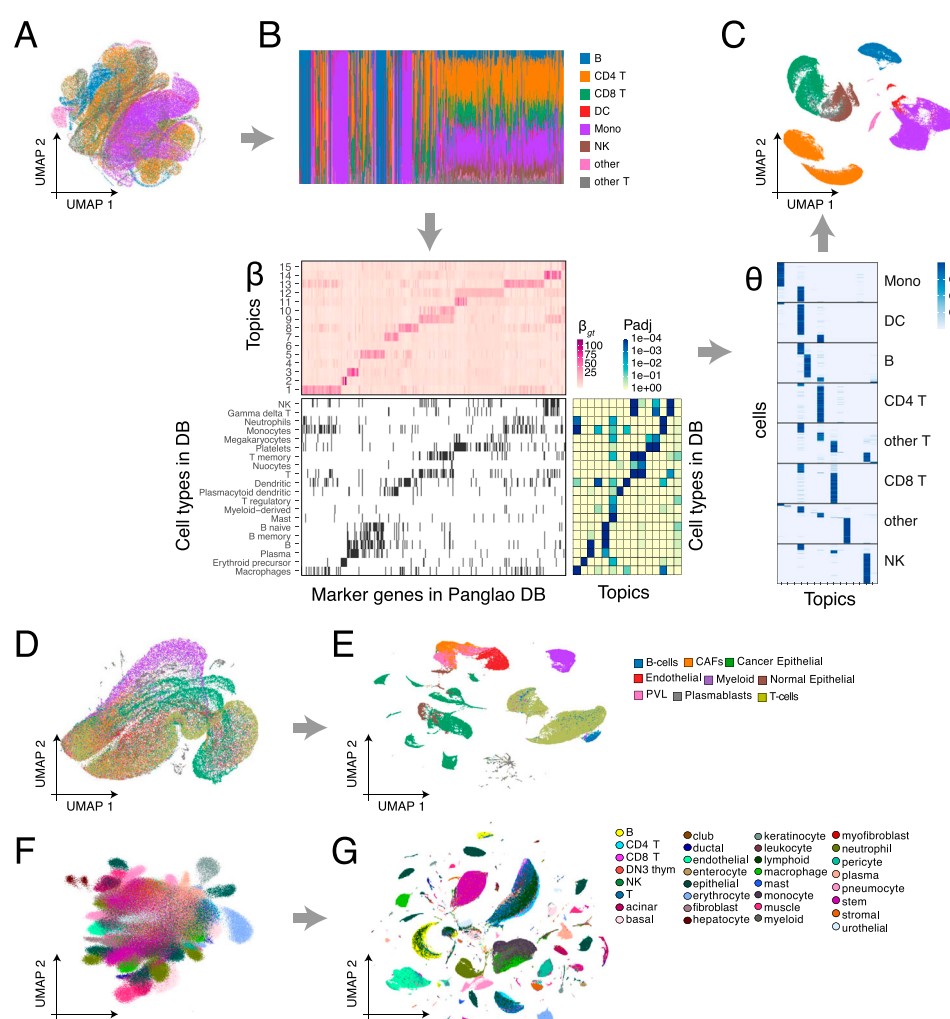

**Figure 4.    ASAP accurately annotates cell types in real-world large-scale data.**
**(A, B, C, D, E, F, G)** *Data sets*: PBMC data (Hao et al, 2021) (A, B, C), breast cancer mammary tissues (Wu et al, 2021) (D, E), and Tabula Sapiens data (F, G) (Tabula Sapiens Consortium et al, 2022). **(A, B, C)** We show the three steps of ASAP applied to the PBMC data (A, B, C). **(A)** Step 1: Uniform Manifold Approximation and Projection (UMAP) drawn directly from the RP results. The colours represent different cell types previously annotated (level 1) by the original authors (Hao et al, 2021). **(B)** Step 2: pseudobulk data (top) along with the Poisson matrix factorization result—a dictionary matrix β (bottom). We compared the gene activities in the β matrix with the known cell-type marker genes (Franzén et al, 2019). **(C)** Step 3: fast cell-level regression analysis to recover topic proportions of all the cells. UMAP (top) drawn from the topic proportion results θ (bottom). **(D)** UMAP drawn directly from the RP results of the breast cancer data (Wu et al, 2021). **(E)** UMAP based on the final topic proportion results of ASAP. The colours represent different cell types based on the original study. Because of the space limit, we show the topic proportion matrix θ in Fig S1. **(F)** UMAP directly drawn from the RP results of the Tabula Sapiens data (Tabula Sapiens Consortium et al, 2022). **(G)** UMAP based on the final topic proportion results of ASAP. The colours represent different cell types based on the original study (the broad level). Because of the space limit, we show the topic proportion matrix θ in Fig S2.

and (3) the first draft of the Tabula Sapiens project (Tabula Sapiens Consortium et al, 2022) (Fig 4F and G), consisting of 483,152 cells and 58,604 genes/transcripts.

ASAP's dictionary matrix β (topic-specific gene expressions) strongly enriches the known cell-type marker genes, PanglaoDB (Franzén et al, 2019), within the same topics (bottom, Fig 4B); hence, the resulting cell topic proportions clearly differentiate cells based on the actual cell-type annotations (Fig 4C). We performed gene set enrichment analysis (GSEA) within each topic by fgsea (Korotkevich et al, 2021 *Preprint*), which provides a faster and more accurate approximation of the traditional GSEA method (Subramanian et al, 2005). Interestingly, the same level of resolution was not observed in the first RP step (Fig 4A). Many PB samples were rather heterogeneous, mixing multiple cell types (top, Fig 4B).

Considering that no label information was given to the factorization step, our intermediate results demonstrate that the non-negative matrix factorization step is indeed important, and the previous PB construction captures necessary information while compressing data complexity drastically. In the other real-world data analysis, we also observed that PB construction provided rough ideas about cell type–specific patterns (Fig 4D and F); the subsequent PMF steps refine topic-specific gene activities to result in high-resolution cell-type annotations in the end (Fig 4E and G). We also noted that the topics do not always agree with the cell types, which are often best characterized by cell type–specific surface markers and transcription factors. For instance, the topics of the breast cancer data may implicate finer resolution of cancer subtypes (Fig 4E), or the different cell types could be strongly affected by shared topic-specific genes (Fig S1). For a large study, such as Tabula Sapiens, cell-type annotations are often manually curated by different researchers with different domain knowledge, for example, dividing the jobs tissues-by-tissue. Nonetheless, we found the original annotations generally correspond to topics (Fig S2) and are well separated in the topic space (Fig 4G).

Moreover, we acknowledge that UMAP is perhaps not an ideal way to show a clear separation of distinct cell types manifested in the high-dimensional gene/feature space. Richer information is captured within the dictionary (β) and topic proportion (θ) matrices. We report supplementary figures to show the resulting topic

proportion matrices (θ, cell by topic in Figs S1 and S2) and the dictionary matrices as the supplementary material.

### Pseudobulk samples correspond to bulk expression samples

We had expected that PB samples would behave similar to cell type–sorted bulk samples. However, the previous real-world PB data showed quite a level of heterogeneity, mixing multiple cell types—especially those closely related cell types, such as CD4$^+$ and CD8$^+$ T cells. Such results also show that RP alone is insufficient to investigate fine-resolution cellular diversities.

To better understand the characteristics of PB samples, we investigated single-nucleus RNA-seq (snRNA-seq) (Eraslan et al, 2022) and bulk RNA-seq data (GTEx Consortium, 2020) profiled by Genotype-Tissue Expression (GTEx) consortium. This snRNA-seq data set provides gene expression vectors on 209,126 cells sampled from 16 individuals across 10 tissue types, for which the GTEx consortium also provides publicly accessible data (GTEx Consortium, 2020).

We first analysed the snRNA-seq matrix with ASAP and obtained its dictionary matrix β (gene by factor). As can be seen in the GSEA against PanglaoDB (Franzén et al, 2019) marker gene sets by the fgsea method (Subramanian et al, 2005; Korotkevich et al, 2021 Preprint), these topics are significantly associated with known cell types (bottom, Fig 5A). We also found that our unsupervised learning can recover the broad cell-type annotations called by the original study (Fig 5B). Interestingly, most cell types are shared across multiple tissues, yet there still exist several tissue-specific cell types, such as multiple epithelial cell types and myocytes, perhaps more specialized in tissue-specific environments.

Using the dictionary matrix β, we projected the bulk RNA-seq data onto the snRNA-seq topic space (Fig 5C and D); then, we overlaid the snRNA-seq PB data on top of the same coordinates. GTEx tissues are generally well separated from one another (Fig 5C), suggesting different cell-type factions as a primary axis of variation. Moreover, we found that the PB samples overlap with these bulk samples to a large degree. Such good coverage compelled the follow-up experiment to assess whether the projected coordinates could be a basis for cell-type deconvolution step. Simply, we were able to query 100 neighbouring snRNA-seq cells for each bulk sample and count the frequency of each cell type within the neighbours.

The neighbouring cell-type fractions can be interpreted as approximate deconvolution profiles (Fig 5E). Unlike marker gene–based deconvolution methods, no prior knowledge was elicited, and reference cell-type profiles were adaptively selected for each bulk sample. Our results suggest several expected patterns. Substantially, higher fractions of myocytes were found in the sun-exposed side of skin and skeletal muscles (brown colours). A special type of alveolar epithelial cells was specifically found in the lung tissues. Immune cells are present in all the tissues but much less than other cell types. The two types of oesophagus tissues, histologically distinct, are markedly different in terms of the cell-type fractions. We found a more diverse mixture in the oesophagus mucosa, containing a much higher fraction of fibroblasts and pericytes. We further investigated whether the cell-type proportion estimates induce visually separable embedding results on the

UMAP space (Fig S3). It is clearly shown that major cell types are clearly distinctive from one another, forming disconnected clusters of cells of each type (the top panel). Subtypes were also visually distinctive but not as clear as the major categories (the bottom panel), implicating room for further improvement in future work.

### Sensitivity to the choice of hyper-parameters

Finally, we tested how the first step of our ASAP framework can be affected by the number of random projections, as the performance of the subsequent steps depends on it. We simulated single-cell data based on pbmc3k data available in the 10x Genomics website (see the Materials and Methods section for the detailed simulation scheme). Here, we fixed the sequencing depth to 10,000 and very little noise level (ρ = 0.025) and assumed the existing cell-type annotations as a gold standard. As expected, an insufficient number of random projection dimensions led to degraded performance (Fig 6A). However, we also found that the performance quickly recovered with sufficiently deep random projection trees above six, which leads to $2^6$ = 64 pseudobulk samples, and no significant performance gain was observed beyond the optimal level. In practice, we found ASAP with a depth of 10 or above generally performs consistently well enough to construct an informative pseudobulk data matrix (≤1,024 samples).

We also noted that the quality of single-cell data summarized by the coverage of non-zero features within each cell could be an important factor. Based on the same simulation scheme, we conducted rather small-scale experiments but with different levels of sequencing depths, that is, the number of reads mapped within each cell, varying from 400 to 10,000 (Fig 6B). To our surprise, the ASAP (PMF with pseudobulk) and PMF (ASAP-full) substantially outperformed across all the conditions. All the methods found a hard time picking up the right cell type–specific patterns with lots of missing values; hence, the scores with a shallow sequencing depth are significantly lower than those with a deep sequencing depth. The drop was rather steep in other NMF-related methods (Liger and scikit) and Scanpy's Leiden clustering, whereas ASAP and PMF methods tend to maintain high accuracy even with little information.

## Discussion

We propose a novel approximation method that can quickly identify topic structures embedded in large-scale single-cell data. Our method builds on the rationale that the distance between two high-dimensional data points can be preserved in the low-dimensional space constructed by random with a high probability (Johnson & Lindenstrauss, 1984; Bingham & Mannila, 2001; Blum, 2006; Dasgupta & Freund, 2008). Based on this foundation, a similar strategy has been used to speed up a clustering procedure (Wan et al, 2020); however, the quality of RP-based clustering analysis demands multiple random projections and clustering steps, and the resolution of clustering results is limited and unable to delineate a subtle difference between similar cell types; hence, the benefit of reduced computation time and peak memory can be

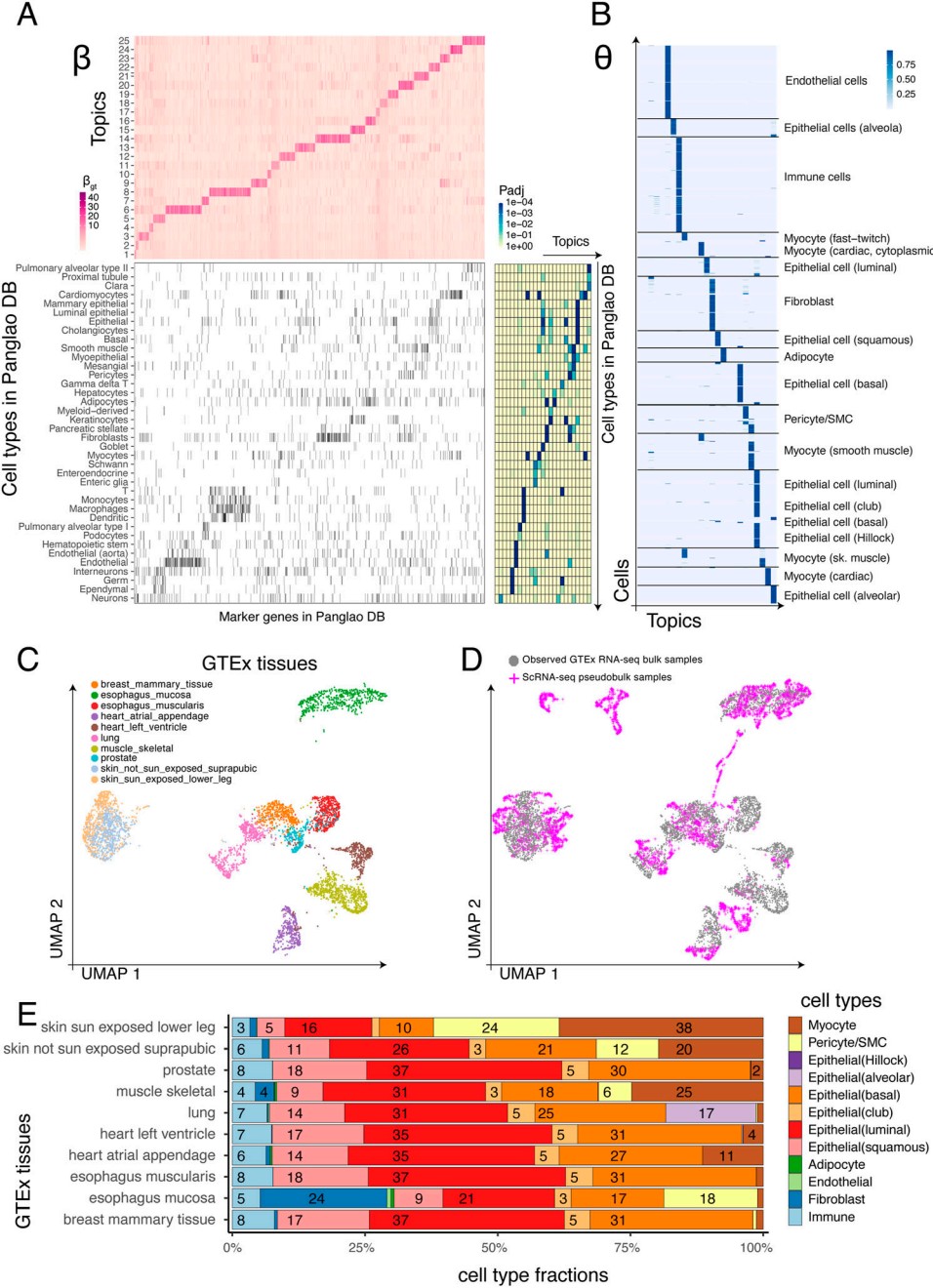

**Figure 5. ASAP deconvolves cell type–specific gene programs in tissue-level data in joint analysis with snRNA-seq data.**

**(D, E)** *Data sets*: Genotype-Tissue Expression project's single-nucleus RNA-seq data (Eraslan et al, 2022) and GTEx project's bulk RNA-seq data (GTEx Consortium, 2020) (D, E). **(A)** Topic-specific gene activity matrix β trained on the pseudobulk data constructed in the GTEx snRNA-seq data (Eraslan et al, 2022) shows strong enrichment of known cell types in PanglaoDB (Franzén et al, 2019). β: topic (rows) by gene (columns). **(B)** Patterns of ASAP's topic proportion estimates θ agree with the original annotations. θ: cell (rows) by topic (columns). **(C)** Uniform Manifold Approximation and Projection embedding of GTEx bulk data, only including tissue types present in the snRNA-seq data. Before the Uniform Manifold Approximation and Projection, we projected the bulk data onto the snRNA-seq topic space. The colours represent different tissues of origin. **(D)** snRNA-seq pseudobulk samples (magenta) are overlaid on the same snRNA-seq topic space where the bulk samples (grey) are projected previously. **(E)** Cell-type proportions estimated by the joint analysis of GTEx snRNA-seq and bulk data. X-axis: cell-type fractions; y-axis: different tissue types; the numbers in the segments: fraction of cell types in percentage.

diminished, and clustering results are often left less interpretable and require further post hoc analysis.

Although our approach begins with random projection steps, which had been previously explored for clustering, our key idea focuses on building a randomly sorted pseudobulk data matrix. We may consider randomly sorted pseudobulk data as a type of "meta"-cell. The potential of meta-cells has been explored in other studies, such as the SEACells algorithm (Persad et al, 2022 *Preprint*), which suggests efficient and accurate archetypal analysis. Unlike noisy (incomplete) cell–cell interaction patterns, meta-cells better represent intrinsic cellular states and complete random drop-out

observations to help ascertain relationships between cis-regulatory regions and target genes. Our work is built on the same premise. Knowing that many cells are stochastic realizations of a "meta-cell," as long as we can construct meaningful aggregates of cells, it is perhaps more sensible to run computational methods on the resulting meta-cell (pseudobulk) data.

In realistic benchmark analysis based on sorted bulk RNA-seq profiles, we demonstrate that learning a topic model on randomly sorted pseudobulk data outperforms unsupervised cell-type annotation tasks. Unlike clustering results, our topic model parameters are immediately interpretable and directly comparable

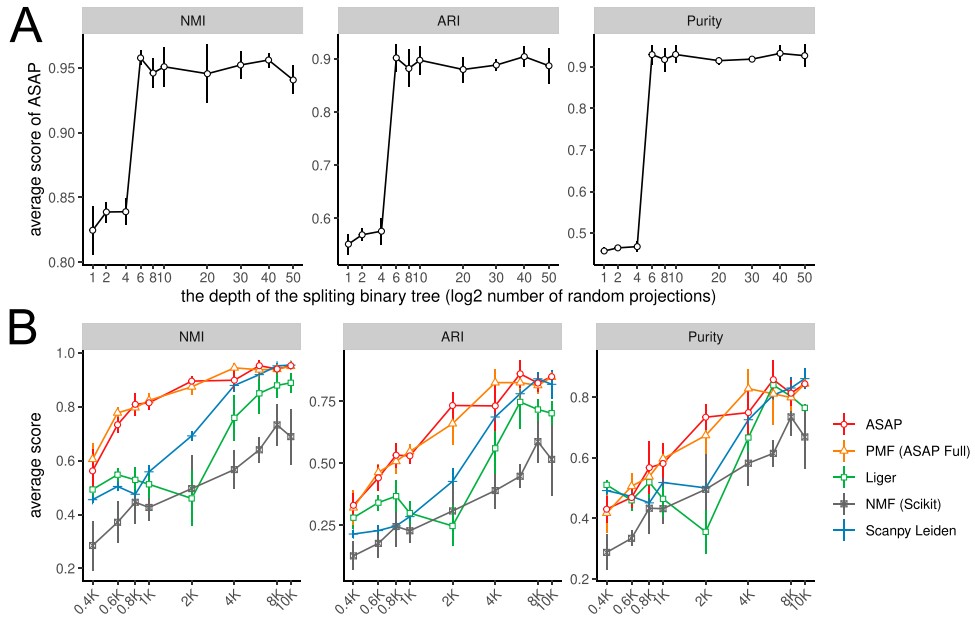

**Figure 6. ASAP with a sufficiently deep binary–sorting tree performs robustly despite shallow sequencing depth.**
**(A)** Varying the number of random projections, we evaluate the accuracy of the clustering patterns resulted from ASAP. X-axis: the depth of random binary trees; y-axis: the accuracy (NMI, ARI, purity). Each experiment was repeated five times. The vertical error bars represent one SD from the mean estimates. NMI, normalized mutual information (Vinh et al, 2010); ARI, adjusted Rand index (Rand, 1971); purity defined in references (Liu et al, 2020; Kriebel & Welch, 2022). For all the metrics, the higher, the better. **(B)** Clustering performance-varying sequencing depth parameters from 400 to 10,000 per cell. We simulated the data using 3,000 PBMCs while sampling five cells with the noise level $\rho$ = 0.025, and resolved clustering by the Leiden method with the resolution parameter 0.1. For ASAP, we set the number of random projections to 10. X-axis: sequencing depth of the simulated data; y-axis: the accuracy (NMI, ARI, purity).

against known marker gene databases. Furthermore, we show that our approach resembles a data-generating scheme of bulk RNA-seq data, and the inference results are easily transferable for deconvolution analysis. Especially for a data set with shallow sequencing depth, our method tends to show robust performance, which is highly desirable for large-scale single-cell data that contain a large fraction of missing observations.

Our approach can be extended in many different ways. We can intuitively apply RP operations to reduce the feature space for epigenomics, genomics, and other types of sparse matrix data. Although aggregating count data within a PB sample generally works robustly, we can combine batch effect adjustment and advanced depth normalization methods to further improve the quality of the resulting PB data. If needed, multi-omics data integration can take place at a PB level followed by nearest neighbour matching (Haghverdi et al, 2018; Hie et al, 2019a; Polański et al, 2020) or even optimal transport (Demetci et al, 2022). Moreover, in the case of heterogeneous multi-modal data integration tasks (Hristov et al, 2022), we expect matching results within the randomly projected space to strike a balance between runtime and theoretical guarantee.

Joint analysis in single-cell omics has gained more attraction; a recent method, such as multi-modal topic modelling (Kazwini & Sanguinetti, 2024), can benefit from our ASAP framework, speeding up Bayesian inference and achieving robustness in the presence of unwanted missing values. Any machine-learning algorithm can be used to learn a topic-specific feature frequency matrix. A deep learning–based generative model can be more efficiently trained by alternatively applying RP and PB steps; proper modelling that incorporates prior knowledge of cell type-specific marker genes and dependency structures will also substantially improve inference results (Elyanow et al, 2020; Townes & Engelhardt, 2023).

# Materials and Methods

## Probabilistic topic modelling by ASAP

### Goal: topic modelling for single-cell data analysis

Single-cell transcriptomic profiling results in a count matrix $X \in R^{D \times n}_{\geq 0}$ of $n$ cells where each column vector lies in the $D$-dimensional space. The dimensionality $D$ corresponds to the number of features, such as genes and non-coding RNA molecules. Each element $X_i$ ($j \in [n]$) quantifies how many reads were mapped onto a gene $i$ within a cell $j$; hence, each value is non-negative by nature and generated by a counting process. Throughout this work, we will indicate a gene by $i \in [D]$, denote a cell/sample by $j$, and use $k \in [K]$ for a topic index. In total, we have $D$ features and $n$ cells, and the goal is to estimate $K$ topics, $(\beta_1, \dots, \beta_K)$, and the corresponding topic proportion vector, $(\theta_1, \dots, \theta_n)$, for each cell.

Given that each cell $j$, the column vector $x_j$, sits on a topic space, $k \in [K]$, we can assign a topic proportion of a topic $k$ to the cell $j$, namely, $\theta_{jk}$ values, to quantify which and how much each topic affected each cell. Defining a matrix $\beta$ with each element $\beta_{ik}$ for an expected expression value for a gene $g$ on a topic $k$, we assume that an observed gene expression matrix $X$ was generated by a linear combination of the two matrices—$\beta$ and $\theta$. More precisely, we have

$$E[X_{ij} | \beta, \theta] = \sum_{k=1}^{K} \beta_{ik} \theta_{jk},$$

where the equality holds proportionally up to some scaling factor. The goal is to estimate the unknown gene topic (program) dictionary ($\beta$) and the topic proportion matrix ($\theta$) provided with an observed gene expression matrix $X$ and the fact that distinctive

gene expression programs/topics exist and are expressed differently according to each cell's topic representation.

### PMF

The PMF model (Gopalan et al, 2014, 2016; Levitin et al, 2019) treats the same non-negative factorization problem as the Bayesian inference problem, assuming that each element of the data matrix, $X_{ij}$, was generated by the following generative model:

1. Sample topic $k$-specific gene/feature expression: $\beta_{ik}$ ~ Gamma($a_0, b_0$).

2. Sample each cell/sample's activity/loading on a topic $k$: $\theta_{jk}$ ~ Gamma($a_0, b_0$).

3. Constitute the sampling rate $\lambda_{ij}$ for gene/feature $i$ in each cell/sample $j$ as a linear combination of the two parameters, namely, $\lambda_{ij} = \Sigma_k \beta_{ik} \theta_{jk}$.

4. Sample gene expression for a gene $i$ in a sample $j$ from the Poisson distribution: $X_{ij}$ ~ Poisson($\lambda_{ij}$).

More precisely, we define the probability density function (PDF) of Gamma($a_0, b_0$) as

$$p(\lambda|a,b) = \frac{b^a}{\Gamma(a)}\lambda^{a-1}\exp(-b\lambda),$$

and the PDF of Poisson($\lambda$) as

$$p(X|\lambda) = \frac{\lambda^X}{X!}\exp(-\lambda).$$

Remark: The scale of the $\beta$ and $\theta$ parameters is arbitrary, and they can adapt to each other, for example, $\beta_k\theta_k^\top = (s_k^{-1}\beta_k)$. The lack of identifiability in scaling factors brings about an interesting equivalence relationship between the PMF and the conventional multinomial topic model likelihood (Carbonetto et al, 2021 *Preprint*). It is straightforward to translate the PMF parameters into the conventional topic model parameters by adjusting the scales of $\beta$ and $\theta$ parameters adaptively, such that $\Sigma_i \beta_{ik} = 1$ and $\Sigma_k \theta_{jk} = 1$.

### Detailed steps of the ASAP algorithm

#### Step 1. Random projection (RP) and pseudobulk (PB) construction

A full $D \times n$ data matrix $X$ can be sparse and too big to be read and densified in local memory. In the first step, each column/cell vector $\mathbf{x}_j$ is compressed to a smaller, manageable $d \times n$ matrix $Q$ by randomly projecting the total data matrix $d$ times onto the $d$-dimensional space ($d \ll D$). For each random projection ($k = 1, \dots, d$), we use isotropic $D$-dimensional Gaussian vector $r_k$ and take the dot product with each cell vector, namely,

$$q_k^\top \leftarrow [r_k^\top x_1, \dots, r_k^\top x_n].$$

To ensure that each RP vector is independent of the others, we perform (economical) singular value decomposition (SVD) of the $Q$ matrix and construct orthogonal $d \times n$ RP matrix $\tilde{Q}$. After applying row-wise standardization, for each cell $j \in [n]$, we can quickly convert $\tilde{q}_j \equiv$ to a vector of $d$ binary codes by setting each $k$-th

coordinate to 1 if the $\tilde{Q}_{kj}$ is positive; otherwise, it is 0 (Fig 1B). This binary code vector can be used as a binary sorting tree, literally converting a binary number to a decimal one, with which we can quickly scatter cells into $2^d$ buckets (leaf nodes). In each bucket/leaf node, we can incrementally aggregate expression vectors to a PB sample, and each PB vector $y_l$ is later normalized to have uniform sequencing depths ($l = 1, \dots, 2^d$). For simplicity, we ignore empty buckets; hence, the resulting PB matrix can have at most $2^d$ columns/samples.

· Construct a raw RP matrix $Q_{d \times n} \leftarrow R_{d \times D}X_{D \times n}$.

· Orthogonalize the $Q$ matrix and build $\tilde{Q}$ by SVD and standardization.

· Make a binary $B$ matrix by setting each element.

$$B_{kj} = \begin{cases} 1 & \tilde{Q}_{kj} > 0 \\ 0 & \text{otherwise} \end{cases},$$

where $k = 1, \dots, d$ and $j = 1, \dots, n$.

· Convert this $B$ matrix to a leaf node (bottom bucket) membership matrix:

$$L_{lj} = \begin{cases} 1 & \sum_{k=1}^{d} 2^{k-1}B_{kj} = l \\ 0 & \text{otherwise} \end{cases},$$

where $k = 1, \dots, d$ and $j = 1, \dots, n$.

· Aggregate cells within each leaf node into a PB sample:

$$y_l = \sum_{j:L_{lj}=1} x_j$$

· Normalize $y_l \leftarrow y_l/\|y\|_1 \times 10^4$.

#### Step 2. Variational inference of the PMF model

Next, we decompose the $d \times L$ PB matrix $Y$ by the PMF model ($L \le 2^d$), modelling each PB data point as the Poisson distribution with the rate parameter that can be decomposed into dictionary and loading matrices (see the previous subsection for details). However, the log-likelihood,

$$F = \sum_{ij} Y_{ij}\log\left(\sum_{k=1}^{K}\beta_{ik}\theta_{jk}\right) - \sum_{ij}\sum_{k=1}^{K}\beta_{ik}\theta_{jk} + \text{constant},$$

is non-conjugate with the Gamma distributions of the $\beta$ and $\theta$ parameters. Following the previous variational inference algorithm (Gopalan et al, 2014, 2016; Levitin et al, 2019), we need to introduce auxiliary variables to construct evidence lower bound (ELBO) under the mean-field variational approximation (using Jansen's inequality):

$$F \ge \sum_{ijk} z_{ijk}Y_{ij}\log\frac{\beta_{ik}\theta_{jk}}{z_{ijk}} - \sum_{ij}\beta_{ik}\theta_{jk} + \text{constant} \overset{\Delta}{=} L.$$

This ELBO now permits the following closed-form update equations for the variational distributions.

· Inference of the local/sample-specific topic proportions:

$$\theta_{jk} \Big| z, \beta \sim \mathrm{Gamma}\left(\theta_{jk} \Big| a_0 + \sum_{i=1}^{D} y_{ij} z_{ijk}, b_0 + \sum_{i=1}^{D} E_q[\beta_{ik}]\right).$$

· Inference of the global/dictionary parameters:

$$\beta_{ik} \Big| z, \theta \sim \mathrm{Gamma}\left(\beta_{ik} \Big| a_0 + \sum_{j=1}^{L} y_{ij} z_{ijk}, b_0 + \sum_{j=1}^{L} E_q[\theta_{jk}]\right).$$

· Inference of the auxiliary variable:

$$\log z_{ijk} \leftarrow E_q\left[\log \frac{\theta_{jk}\beta_{ik}}{\sum_k \theta_{jk}\beta_{ik}}\right] + constant, \text{ such that } \sum_{k=1}^{K} z_{ijk} = 1.$$

### Step 3. Prediction using latent variable factorization

Finally, given that we have the expected dictionary matrix, namely, $E[\beta]$ and $E[log\beta]$, we revisit full single-cell data and recover cell-level topic proportion vectors, namely, $\theta_j$ = for each cell $j$ 1, …, $n$. We could use the same variational inference algorithm, only skipping the update for the $\beta$ parameters, but it would involve full-scale inference of the auxiliary variables and not be scalable in practice. Instead, we form a different approximation of the ELBO and calibrate the $\theta$ parameters by solving a massive array of regression problems:

$$L \geq \sum_{j,k}\sum_{i} Y_{ij} t_{jk} \log \frac{\beta_{ik}\theta_{jk}}{t_{jk}} - \sum_{i,j,k} \beta_{ik}\theta_{jk},$$

where $\sum_k t_{jk} = 1$ for all $j$. Letting $\rho_{jk} \triangleq E[t_{jk}]$ under the same variational approximation, we can derive two closed-form update equations. Firstly, we have

$$\log \rho_{jk} = \frac{\sum_i Y_{ij} E_q[\ln \beta_{ik}]}{\sum_i Y_{ij}} + E_q[\ln \theta_{jk}] + constant,$$

where $\sum_k \rho_{jk} = 1$, and have the variational

$$\theta_{jk} \Big| \rho, \beta \sim \mathrm{Gamma}\left(\theta_{jk} \Big| a_0 + \rho_{jk} \sum_{i=1}^{D} y_{ij}, b_0 + \sum_{i=1}^{D} E_q[\beta_{ik}]\right).$$

### Simulation

### Simulation scheme

We generated a single-cell data set from cell type–sorted bulk RNA-seq reference data from the DICE (Database of Immune Cell Expression, Expression quantitative trait loci and Epigenomics) project (Schmiedel et al, 2018) to evaluate the performance of ASAP and other methods with gold standard annotations. We considered the population of 13 cell types, including CD4$^+$ T cells, CD8$^+$ T cells, NK cells, B cells, and monocytes. We use a well-

established data-generating framework implemented in the scDesign2 package—a simulation method based on a Gaussian copula-based sampling scheme (Sun et al, 2021). Gene–gene correlation structures are accurately captured in data simulated by a copula method, and the non-parametric nature of its density estimation step is more suitable for simulation experiments with little modelling assumption. Mathematical details can be found in the original work (Sun et al, 2021), but briefly, we simulated our data as follows.

For each cell type $t$, we generate a single-cell data matrix $Y^{(t)}$ based on a continuous version of the transformed data as follows:

1. Estimate gene-level mean $\mu$ and covariance between genes $\Sigma$ for each cell type $t$.

2. For each cell $j$, we generate bootstrapped copula $y_j$:

– Sample a Gaussian vector $z_j \sim N(\mu_t, \Sigma_t)$.
– Construct a stochastic version of the reference gene expression vector $s_j$ by sampling with replacement.
– Sort the bootstrapped gene expressions $S_{gj}$, that is, $S_{(1)j} < S_{(2)j} < \cdots < S_{(D)j}$.
– Assign each gene $g$'s gene expression $Y_{gj}$ in two steps:

· Identify the ascending order $r$ for $Z_{gj}$
· Take the bootstrapped $S_{(r)j}$ and set $Y_{gj}$ to this value.

3. Repeat the above steps until we sample the desired number of cells.

Next, we simulated a separate single-cell data matrix from a null model that captured the global gene expression pattern across all cell types in the bulk samples. To account for the background and cell type–invariant patterns, we generate the null data $\tilde{Y}$ by sampling a gene vector $\tilde{y}$ from a multinomial distribution with the empirical gene frequencies, ignoring cell-type identities. Depending on the $\rho \in (0,1)$ value, we differently mix the cell type–specific foreground and the background signals: $y_j \leftarrow y_j^{(t)}\rho + \tilde{y}_j(1-\rho)$. Both foreground and background data were normalized to have the same sequencing depth.

## Data Availability

We used single-cell RNA-seq data made publicly available from the original works: Hao et al (2021): https://azimuth.hubmapconsortium.org/; Wu et al (2021): GSE176078; Eraslan et al (2022): https://singlecell.broadinstitute.org/single_cell/study/SCP1479; 10x Genomics: https://cf.10xgenomics.com/samples/cell/pbmc3k/pbmc3k_filtered_gene_bc_matrices.tar.gz; and Tabula Sapiens: https://tabula-sapiens-portal.ds.czbiohub.org/. We also share our Python package code: https://github.com/causalpathlab/asapp.

## Supplementary Information

# Acknowledgements

This work was supported by the BC Cancer Foundation, NSERC Discovery, and Canada Research Chair Tier-2 (YP Park). We also acknowledge support from the UBC Four Year Fellowship (S Subedi).

## Author Contributions

S Subedi: conceptualization, resources, data curation, software, formal analysis, funding acquisition, investigation, visualization, methodology, and writing—original draft, review, and editing.
TS Sumida: conceptualization, investigation, methodology, and writing—original draft.
YP Park: conceptualization, resources, data curation, software, formal analysis, supervision, funding acquisition, investigation, visualization, methodology, project administration, and writing—original draft, review, and editing.

## Conflict of Interest Statement

The authors declare that they have no conflict of interest.

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
