## [Reviewer comments · Life Science Alliance]

Life Science Alliance

A scalable approach to topic modelling in single-cell data by approximate pseudobulk projection

Sishir Subedi, Tomokazu Sumida, and Yongjin Park

DOI: <https://doi.org/10.26508/lsa.202402713>

Corresponding author(s): Yongjin Park, University of British Columbia

Review Timeline:

Submission Date:	2024-03-12
Editorial Decision:	2024-05-28
Revision Received:	2024-07-01
Editorial Decision:	2024-07-23
Revision Received:	2024-07-29
Accepted:	2024-07-30

Transaction Report:

May 28, 2024

Re: Life Science Alliance manuscript #LSA-2024-02713-T

Yongjin Park
The University of British Columbia

Dear Dr. Park,

Thank you for submitting your manuscript entitled "A highly scalable approach to topic modelling in single-cell data by approximate pseudobulk projection" to Life Science Alliance. The manuscript was assessed by expert reviewers, whose comments are appended to this letter. We invite you to submit a revised manuscript addressing the Reviewer comments.

Thank you for this interesting contribution to Life Science Alliance. We are looking forward to receiving your revised manuscript.

Sincerely,

B. MANUSCRIPT ORGANIZATION AND FORMATTING:

Reviewer #1 (Comments to the Authors (Required)):

The paper presents an algorithmic approach to scale up topic modelling (in its equivalent Poisson Matrix Factorisation, PMF, version) to large single-cell sequencing data sets. The core idea of the approach is to preface the algorithm with a random-projection which enables the creation of a fast pseudo-bulk, which is then fed to the PMF algorithm. The authors evaluate this idea both in terms of accuracy and computational efficiency on a simulation study and a number of data sets.

The claim that the approach is computationally more efficient than just applying PMF to real data is well substantiated, although one would be surprised if that were not the case. The claim on the accuracy is overall fairly convincing, but there are some critical aspects that should be explored further. In particular, I think it would be essential in the simulation to vary the coverage; given that many very large scRNA-seq data sets are extremely sparse, the risk that a random direction might be uninformative cannot be discounted.

One additional issue that I did not find discussed is the tuning of the hyper parameters, namely the number of topics and the number of directions in the random projection. If the algorithm is sensitive to these choices and requires additional tuning to determine the hyper parameters, then its computational advantages might be diluted.

Finally, I notice a very recent reference of the usage of topic models for single cell multi-omics, <https://link.springer.com/article/10.1186/s13059-024-03180-3>, which the authors might want to include to further demonstrate the interest in this type of models in the single-cell analysis community.

Reviewer #2 (Comments to the Authors (Required)):

The authors present a new method for uncovering single cell data topics that uses much less computational resources than other methods and achieves comparable performance in downstream tasks such as clustering. The key idea behind ASAP (the name of the tool) is to first use random projections of the huge and very sparse single cell expression matrix to a lower dimensional space and employ a basic binary tree to aggregate the projected single cells vectors into a pseudo-bulk data matrix. This pseudo-bulk matrix is then factorized into topics and loadings matrices using Poisson Matrix Factorization (essentially NMF) that can be used to deconvolve the cell-level topic proportions.

I think that the authors' idea of employing as a first step random projections coupled with a binary tree is interesting. Given that the size of current single-cell experiments is massive and increasing, tools that can scale and run using less resources are needed. Overall, I find the paper to be well written, however, I have several major concerns about the evaluation of the tool.

Major remarks:

Comparison to other basic methods is shown only for simulated data in Figure 2. It will greatly boost the claim that ASAP works well if the same comparison is done on real data (i.e in Figure 3)

I did not see what the size of the dimension of the random projection is anywhere in the text. How is this key parameter set at all? Is it the same across all datasets? Is this reasonable and what experiments have being carried to justify it

While comparing the visual of UMAP using the random projections from step1 of the algorithm to UMAP using the topics metrics (at the end of step3 of the algorithm) is very helpful in establishing that steps2 and 3 are useful and yield a better embedding of the data, these plots start to lose value in Figure 4. Here, showing how some other approaches perform could provide further lines of evidence that ASAP's topics work in practice.

Explain in the caption of Figure 3 as well as in the text why the curves for PFM and NMF do not go to 100,000 cells

Discussion is not very informative, should have more citations (i.e doi 10.1101/2022.04.02.486748) and include some discussion how other tools (i.e DOI: 10.1093/bioinformatics/btac481) that rely on ad hoc clustering to scale to larger single cell sets can potentially benefit from ASAP's topics.

I did not find a link to where the code is (i.e github page)

Minor:

Line 5: the relationship is reversed: computational methods accompany the sequencing technology, not the vice versa

Line 281: "\rho="

Line 234: uncompiled reference "???"

Reviewer #1

The paper presents an algorithmic approach to scale up topic modelling (in its equivalent Poisson Matrix Factorisation, PMF, version) to large single-cell sequencing data sets. The core idea of the approach is to preface the algorithm with a random-projection which enables the creation of a fast pseudo-bulk, which is then fed to the PMF algorithm. The authors evaluate this idea both in terms of accuracy and computational efficiency on a simulation study and a number of data sets.

We appreciate the reviewer's comments. Here, we make our best effort to address all the points raised in the following texts and make proper adjustments in the main texts.

1.1. Scalability by approximation

The claim that the approach is computationally more efficient than just applying PMF to real data is well substantiated, although one would be surprised if that were not the case.

One of the main goals of this paper is to show that an intuitive algorithmic approach can expedite model fitting and achieve scalability in memory management in single-cell genomic analysis. The paper was also motivated by seeing how such an intuitive idea had yet to be adopted in research. We hope similar types of ideas could help train more sophisticated AI/ML models as a data augmentation strategy.

1.2. Hyperparameter tuning

1.2.1. Sequencing depth (the coverage of genes/features) ASAP (and PMF) preformed robustly with Shallow sequencing depth.

The claim on the accuracy is overall fairly convincing, but there are some critical aspects that should be explored further. In particular, I think it would be essential in the simulation to vary the coverage; given that many very large scRNA-seq data sets are extremely sparse, the risk that a random direction might be uninformative cannot be discounted.

We agree with the reviewer's comment. It is an interesting and important aspect that we have missed. We conducted new sets of benchmark experiments to gauge the sensitivity and robustness of the methods, varying the depth parameters. Assuming genes are uniformly randomly sampled, a prescribed sequencing depth parameter directly control the number of genes/features observed vs. randomly dropped. We included the following figure in the main text and shared the results as supplementary material.

Figure 1: **ASAP with shallow sequencing depth.** Each experiment was repeated five times. The vertical errorbars represent one standard deviation from the mean estimates. NMI: normalized mutual information (Vinh, Epps, and Bailey 2010); ARI: adjusted Rand index (Rand 1971); purity defined in (Liu et al. 2020; Kriebel and Welch 2022). For all the metrics, the higher, the better.

Overall, we found PMF methods (ASAP and ASAP-full) perform significantly more robustly than the other factorization (e.g., `liger`) and graph-based clustering (`leiden` in `scanpy`) methods. There are several reasons why the existing methods are particularly vulnerable to shallow coverage: (1) Feature and cell selection steps can be misguided after normalization; (2) The objective functions, such as least square and modularity, of clustering and factorization steps can fluctuate greatly in the presence of outliers and missing values.

1.2.2. The number of random projections

ASAP performs well with sufficiently many random projection dimensions.

One additional issue that I did not find discussed is the tuning of the hyper parameters, namely the number of topics and the number of directions in the random projection. If the algorithm is sensitive to these choices and requires additional tuning to determine the hyper parameters, then its computational advantages might be diluted.

We appreciate the reviewer’s careful assessment. This is another important point that we have missed. Including sensitivity analysis of hyper-parameters, we were able to improve our paper and make our discussions richer. We revised the main text accordingly.

As can be seen in the above panels, creating a pseudo-bulk matrix with sufficiently many columns is necessary for the subsequent PMF method to excavate an informative dictionary matrix. For many single-cell RNA-seq data analysis, we found setting the tree depth greater than six was sufficient enough while achieving substantially faster runtime performance. In all the experiments in this work, we set the depth to ten, which then leads to at most 1,024 pseudo-bulk samples.

1.3. Extension of ASAP

Finally, I notice a very recent reference of the usage of topic models for single cell multi-omics, <https://link.springer.com/article/10.1186/s13059-024-03180-3>, which the authors might want to include to further demonstrate the interest in this type of models in the single-cell analysis community.

Figure 2: **ASAP needs sufficiently many binary sorting steps.** Each experiment was repeated five times. The vertical errorbars represent one standard deviation from the mean estimates. NMI: normalized mutual information (Vinh, Epps, and Bailey 2010); ARI: adjusted Rand index (Rand 1971); purity defined in (Liu et al. 2020; Kriebel and Welch 2022). For all the metrics, the higher, the better.

We read this paper (Kazwini and Sanguinetti 2024) with great interest. The authors propose a dual-headed probabilistic topic model for multi-modal data integration of scATAC and scRNA-seq data. The premise is that epigenetic and transcriptomic expressions unanimously suggest shared cellular topics. Multiomics data integration is an immediate future direction that we are currently pursuing. We agree with the reviewer’s comment that we can introduce pseudobulk approximation steps to expedite multi-modal topic model estimation algorithms. We cited the paper and discussed it in future directions.

Reviewer #2

The authors present a new method for uncovering single cell data topics that uses much less computational resources than other methods and achieves comparable performance in downstream tasks such as clustering. The key idea behind ASAP (the name of the tool) is to first use random projections of the huge and very sparse single cell expression matrix to a lower dimensional space and employ a basic binary tree to aggregate the projected single cells vectors into a pseudo-bulk data matrix. This pseudo-bulk matrix is then factorized into topics and loadings matrices using Poisson Matrix Factorization (essentially NMF) that can be used to deconvolve the cell-level topic proportions.

We thank the reviewer for reading through our work and accurately summarizing the basic ideas behind our method.

I think that the authors’ idea of employing as a first step random projections coupled with a binary tree is interesting. Given that the size of current single-cell experiments is massive and increasing, tools that can scale and run using less resources are needed.

We appreciate the reviewer for finding value in our work. The project spun out of our frustration with memory shortage and runtime issues. At the same time, we did not want to compromise the original feature space by applying an arbitrary feature selection method. Running a computational pipeline in single-cell data analysis is still a bottleneck for many researchers and practitioners, especially those without necessary computational resources, such as high-memory GPU machines.

Overall, I find the paper to be well written, however, I have several major concerns about the evaluation of the tool.

We tried our best to revise manuscript while addressing the reviewer’s comments. We appreciate the reviewer’s comments.

2.1. Performance in observed datasets

Comparison to other basic methods is shown only for simulated data in Figure 2. It will greatly boost the claim that ASAP works well if the same comparison is done on real data (i.e in Figure 3)

We wanted to evaluate clustering performance on realistic simulated data with clear gold standards established by experimental procedures, such as FACS (Fluorescence-activated cell sorting), found such one (Schmiedel et al. 2018) and simulated single-cell data matrices while preserving gene-gene dependency structures by Gaussian copula.

We respectfully argue that many seemingly objective single-cell benchmark data are not so adequate for this purpose since their cell type labels were manually annotated based on graph-based clustering methods implemented in **Seurat** or **Scanpy**, making graph-based methods look highly accurate and less prone to over-fitting than other ML/AI methods with much fewer parameters. This is why we need a realistic simulation method, e.g., **scDesign** (Sun et al. 2021).

We ask readers to be cautious when interpreting the results on observational datasets labelled by a computational pipeline (e.g., feature selection, PCA followed by Leiden clustering). Higher performance may implicate a higher level of overlap with the initial computational analysis. Nonetheless, we measured clustering performance on the three large-scale datasets and reported NMI (normalized mutual information) here:

- PBMC: Scanpy Leiden 0.75, ASAP 0.72, Liger 0.69
- BRCA: Scanpy Leiden 0.79, Liger 0.67, ASAP 0.67
- Tabula Sapiens: Scanpy Leiden 0.85, ASAP 0.75, Liger 0.68

Other metrics show the same pattern. We tweaked the resolution parameter (0.1) of the **leiden** clustering steps to achieve the best possible performance. We found a similar level of agreement of **liger** and **ASAP** results with the original cell type labels.

2.2. The number of random projections

I did not see what the size of the dimension of the random projection is anywhere in the text. How is this key parameter set at all? Is it the same across all datasets? Is this reasonable and what experiments have being carried to justify it

Since the other reviewer asked the same question, we refer the review to the previous Section 1.2.1. and 1.2.2. We acknowledge that this is crucially important and tested our method by conducting additional computational experiments. Briefly, ASAP, with a sufficiently deep tree depth parameter set, outperforms other methods in our simulation experiments and was affected little by the hyperparameter tuning. We included additional results and discussions in the main texts.

While comparing the visual of UMAP using the random projections from step1 of the algorithm to UMAP using the topics metrics (at the end of step3 of the algorithm) is very helpful in establishing that steps2 and 3 are useful and yield a better embedding of the data, these plots start to lose value in Figure 4. Here, showing how some other approaches perform could provide further lines of evidence that ASAP's topics work in practice.

We showed UMAP to provide a quick, intuitive view of overall performance. We agree with the reviewer's assessment that UMAP starts to lose value as the sample size of the data sets increases. This is why we included supplementary figures to show the resulting topic proportion matrices (cell by topic). Due to limited space, we simply added UMAP figures. We revised the texts to help the readers refer to the supplementary figures.

2.4. Missing simulation experiment results

Explain in the caption of Figure 3 as well as in the text why the curves for PFM and NMF do not go to 100,000 cells

We revised the figure legend and main text to clarify why experiments were not done on certain conditions: For some methods, our computing resources were not big enough to conduct a full matrix factorization with a full set of features. When we conducted our experiments, we found that `liger` was implemented carefully enough to handle large memory issues.

2.5. Enriched discussions

Discussion is not very informative, should have more citations (i.e doi 10.1101/2022.04.02.486748) and include some discussion how other tools (i.e DOI: 10.1093/bioinformatics/btac481) that rely on ad hoc clustering to scale to larger single cell sets can potentially benefit from ASAP's topics.

Thank you so much for pointing out these articles. Yes, they are very relevant. We believe there can be fruitful discussions. There were some similarity in concept but the way the ideas was implemented differs from our work. It would be very interesting to discuss technical aspects in detail. We cited these papers (Persad et al. 2022; Hristov, Bilmes, and Noble 2022) and discussed them in future directions.

2.6. Missing github link

I did not find a link to where the code is (i.e github page)

We apologize for our oversight. We made the GitHub repository public now.

2.7. Typos

Line 5: the relationship is reversed: computational methods accompany the sequencing technology, not the vice verse

Line 281: "="

Line 234: uncompiled reference "???"

We fixed these typos. Thank you.

References

- Hristov, Borislav H, Jeffrey A Bilmes, and William Stafford Noble. 2022. “Linking Cells Across Single-Cell Modalities by Synergistic Matching of Neighborhood Structure.” *Bioinformatics* 38 (Suppl_2): ii148–ii154.
- Kazwini, Nour El, and Guido Sanguinetti. 2024. “SHARE-Topic: Bayesian Interpretable Modeling of Single-Cell Multi-Omic Data.” *Genome Biol.* 25 (1): 55.
- Kriebel, April R, and Joshua D Welch. 2022. “UINMF Performs Mosaic Integration of Single-Cell Multi-Omic Datasets Using Nonnegative Matrix Factorization.” *Nat. Commun.* 13 (1): 780.
- Liu, Baolin, Chenwei Li, Ziyi Li, Dongfang Wang, Xianwen Ren, and Zemin Zhang. 2020. “An Entropy-Based Metric for Assessing the Purity of Single Cell Populations.” *Nat. Commun.* 11 (1): 3155.
- Persad, Sitara, Zi-Ning Choo, Christine Dien, Ignas Masilionis, Ronan Chaligné, Tal Nawy, Chrysothemis C Brown, Itsik Pe’er, Manu Setty, and Dana Pe’er. 2022. “SEACells: Inference of Transcriptional and Epigenomic Cellular States from Single-Cell Genomics Data.” *bioRxiv*.
- Rand, William M. 1971. “Objective Criteria for the Evaluation of Clustering Methods.” *J. Am. Stat. Assoc.* 66 (336): 846–50.
- Schmiedel, Benjamin J, Divya Singh, Ariel Madrigal, Alan G Valdovino-Gonzalez, Brandie M White, Jose Zapardiel-Gonzalo, Brendan Ha, et al. 2018. “Impact of Genetic Polymorphisms on Human Immune Cell Gene Expression.” *Cell*, November.
- Sun, Tianyi, Dongyuan Song, Wei Vivian Li, and Jingyi Jessica Li. 2021. “ScDesign2: A Transparent Simulator That Generates High-Fidelity Single-Cell Gene Expression Count Data with Gene Correlations Captured.” *Genome Biol.* 22 (1): 163.
- Vinh, Nguyen Xuan, Julien Epps, and James Bailey. 2010. “Information Theoretic Measures for Clusterings Comparison: Variants, Properties, Normalization and Correction for Chance.” *J. Mach. Learn. Res.* 11 (95): 2837–54.

July 23, 2024

RE: Life Science Alliance Manuscript #LSA-2024-02713-TR

Dr. Yongjin P Park
University of British Columbia
Pathology and Laboratory Medicine, Statistics
675 W 10th Ave
4.108
Vancouver, BC V5Z 1L3
Canada

Dear Dr. Park,

Thank you for submitting your revised manuscript entitled "A scalable approach to topic modelling in single-cell data by approximate pseudobulk projection". We would be happy to publish your paper in Life Science Alliance pending final revisions necessary to meet our formatting guidelines.

- please address Reviewer 2's remaining minor points
- please be sure that the authorship listing and order is correct
- please remove figures from the manuscript file and leave them uploaded separately
- please add your main and supplementary figure legends to the main manuscript text after the references section
- we encourage you to revise the figure legend for Figure 2 such that the figure panels are introduced in alphabetical order
- please add callouts for Figures 2D and S3 to your main manuscript text
- please add an Author Contributions section to your main manuscript text
- please add a Conflict of Interest statement to your main manuscript text
- please include access information to the GitHub site for code access in a Data Availability statement

LSA now encourages authors to provide a 30-60 second video where the study is briefly explained. We will use these videos on social media to promote the published paper and the presenting author (for examples, see <https://docs.google.com/document/d/1-UWCfbE4pGcDdcgzcmiuJl2XMBJnxKYeqRvLLrLS08s/edit?usp=sharing>). Corresponding or first-authors are welcome to submit the video. Please submit only one video per manuscript. The video can be emailed to contact@life-science-alliance.org

A. FINAL FILES:

B. MANUSCRIPT ORGANIZATION AND FORMATTING:

Sincerely,

Reviewer #2 (Comments to the Authors (Required)):

The authors have successfully and fully addressed my concerns. They have improved the rigor of their manuscript by providing hyperparameter tuning and expanded the results section. Just a few minor remarks:

Line 16: "improves" rather than "improve"

Line 102: Second time in a row the sentence starts with "More recently,"

Figure 1 C) matrix factorization method decomposes rather than decompose

July 30, 2024

RE: Life Science Alliance Manuscript #LSA-2024-02713-TRR

Dr. Yongjin P Park
University of British Columbia
Pathology and Laboratory Medicine, Statistics
675 W 10th Ave
4.108
Vancouver, BC V5Z 1L3
Canada

Dear Dr. Park,

Thank you for submitting your Methods entitled "A scalable approach to topic modelling in single-cell data by approximate pseudobulk projection". It is a pleasure to let you know that your manuscript is now accepted for publication in Life Science Alliance. Congratulations on this interesting work.

DISTRIBUTION OF MATERIALS:

Again, congratulations on a very nice paper. I hope you found the review process to be constructive and are pleased with how the manuscript was handled editorially. We look forward to future exciting submissions from your lab.

Sincerely,
